# An Experimental Investigation of the Mechanical Performance of EPS Foam Core Sandwich Composites Used in Surfboard Design

**DOI:** 10.3390/polym15122703

**Published:** 2023-06-16

**Authors:** Sam Crameri, Filip Stojcevski, Clara Usma-Mansfield

**Affiliations:** 1School of Engineering, Deakin University, Geelong, VIC 3216, Australia; 2Institute for Frontier Materials, Deakin University, Geelong, VIC 3216, Australia

**Keywords:** surfing, surf engineering, surfboards, EPS, sandwich composites, thin-walled structures, E-glass, carbon fibre, PET

## Abstract

Surfboard manufacturing has begun to utilise Expanded Polystyrene as a core material; however, surf literature relatively ignores this material. This manuscript investigates the mechanical behaviour of Expanded Polystyrene (EPS) sandwich composites. An epoxy resin matrix was used to manufacture ten sandwich-structured composite panels with varying fabric reinforcements (carbon fibre, glass fibre, PET) and two foam densities. The flexural, shear, fracture, and tensile properties were subsequently compared. Under common flexural loading, all composites failed via compression of the core, which is known in surfing terms as creasing. However, crack propagation tests indicated a sudden brittle failure in the E-glass and carbon fibre facings and progressive plastic deformation for the recycled polyethylene terephthalate facings. Testing showed that higher foam density increased the flex and fracture mechanical properties of composites. Overall, the plain weave carbon fibre presented the highest strength composite facing, while the single layer of E-glass was the lowest strength composite. Interestingly, the double-bias weave carbon fibre with a lower-density foam core presented similar stiffness behaviour to standard E-glass surfboard materials. The double-biased carbon also improved the flexural strength (+17%), material toughness (+107%), and fracture toughness (+156%) of the composite compared to E-glass. These findings indicate surfboard manufacturers can utilise this carbon weave pattern to produce surfboards with equal flex behaviour, lower weight and improved resistance to damage in regular loading.

## 1. Introduction

Sandwich-structured composites are well-established material configurations used for engineering and everyday applications [1,2,3,4]. Typically, sandwich composites consist of a lightweight core material embedded between two thin, stiff facings. These faces are designed to improve the overall material strength in bending and in-plane loads, while the core reduces the weight and maintains the structural integrity of the overall part. Fibre-reinforced polymers (FRPs) are often used as the facing material of sandwich composites due to their excellent mechanical properties, environmental resistances, and low weight [5]. FRPs are classified as a base/matrix material that binds a synthetic or natural filler material into a structure. The most common synthetic filler material is glass fibre due to its relatively low cost compared to other fibres with higher mechanical properties, such as carbon and kevlar. However, issues with biodegradability and recycling have seen natural options such as jute or flax gain popularity [6]. The diverse materials available for FRPs mean the applications of these materials are similarly diverse, including use in automotive, structural, aerospace, marine, and biomedical fields. The core materials used for sandwich composites are also diverse and can range from plastic and metal foams to cell structures (honeycomb, truss, or lattice) or natural materials, such as balsa. The variety of configurations possible with sandwich composites has led many industries to adopt them over traditional materials, such as metals and monolithic FRPs, utilizing their cost and weight saving [7].

Of sandwich-structured composite applications, sporting equipment offers unique opportunities for innovative material research. The scientific interest in sports stems from the desire to achieve peak competitive performance and due to their advantages, such as a high strength-to-weight ratio, composites are commonly used for sports equipment. In 2020, sports and leisure products were the third highest application for carbon fibre-reinforced polymers (CFRP) with an estimated 15 kt, which was only surpassed by the aerospace and wind turbine industries. However, high-performance composite materials are complicated to recycle, and most CFRP waste globally is either sent to landfill or incinerated [8]. Hence, green materials/manufacturing has become a key topic of scientific sports literature due to the global sustainability movement. As such, sports engineers are increasingly focusing on finding a balanced approach to their equipment’s economic, ethical, and mechanical success [9,10,11,12].

Understanding the current scope of sandwich-structured composites specific to sporting goods markets can aid in developing sustainable alternatives to maintain and improve the current performance levels [13]. While many high-performance sports are extremely data-rich, the contemporary sport of surfing lacks benchmark datasets for material performance. A review conducted in 2021 by Romanin et al. revealed that only seventeen articles investigate surfing equipment, with only three focusing on the surfboard directly [14]. Both the novel nature of surf engineering and the relative oversight of the surfboard within this niche is surprising considering the global surfing industry was valued at USD 3.88 billion in 2020, with surfboard sales accounting for 68% of the revenue within surfing [15]. Expanding the current research into the manufacturing and mechanical performance of surfboard sandwich composites is required to provide designers and surfers with quantitative performance metrics to compare and optimise designs. The broader engineering community may also benefit from the expansion of surf literature, as many parallels exist between surfboards, larger marine vessels, and wind turbines [4,16,17]. 

Despite the limited engineering scrutiny, surfboards have undergone significant changes in material composition, paralleling the sport’s global expansion. Traditional solid wood boards built in BCE Hawaii and Polynesia have been superseded by modern sandwich-structured composites, due to the lower weight and improved manoeuvrability of composites while surfing and their rapid production times [18]. The most prevalent surfboard constructions utilise a polyurethane (PU) foam core reinforced by a thin wooden “stringer” through the centreline (to add longitudinal stiffness). Typically, this foam core is sandwiched with a single outer layer of 4oz E-glass with a polyester (PE) resin matrix (Figure 1). This skin is manufactured through artisan hand layup processes known in the industry as “glassing”. The material surfboard composition of PU core and E-glass/PE resin was investigated by Manning, Croksy, and Bandyopadhyay. Their experimental study investigated the flexural behaviour and impact strength of a range of typical PU foam surfboard constructions, with and without stringer reinforcement, and a variety of facing materials [19]. Using a four-point bend test setup, the most common form of critical failure in surfboards known as “creasing” was achieved. Creasing simply refers to the wrinkling of the surface layer of the sandwich composite across the compression face. 

The E-glass/polyester matrix skin achieved the highest maximum flexural load of the stringer-less constructions tested. However, this construction was the most susceptible to impact damage. The impact resistance of the panels was improved by using an S-glass fabric skin and further increased using epoxy resin rather than PE. Interestingly, the authors conclude there is no significant improvement in the flexural properties by changing the facing materials due to the flexural strength being dominated by the properties of the foam. This result is partially supported by a later study from Johnstone, who found minimal flexural strength difference when changing the resin matrix from polyester to epoxy for PU foam/E-glass sandwich composites [20]. However, Johnstone found the fabric reinforcement to be critical to a sandwich composite’s performance, as testing flax, hemp, and bamboo facings revealed a significant reduction in flexural strength but greater ductility when compared to E-glass facing. Johnstone also concluded the core material had the most significant influence on the composite’s strength. This was found through a comparison between bio-foam and PU foam sandwich composites with identical E-glass/PE facings, where the bio-foam had considerably lower flexural strength compared to the standard materials. 

As aforementioned, a spotlight on the global composite waste issues has spurred sports researchers towards sustainable materials, and this trend is also seen in surf research. A study by Shultz determined that a standard PU shortboard will produce approximately 272 kg of emitted CO_2_ over its entire life cycle, 170 kg in production alone [21]. As it is not uncommon for regular recreational surfers to purchase two to three new boards each year, the short functional life of PU boards and the rapid changes in board trends have caused concerns to arise with the growth of the sport [22]. In response, much of the available surfboard material literature is heavily focused on reviewing composites with sustainable fabric reinforcements [20,22,23] and foam replacements [24,25]. Despite the importance of sustainable materials, the strength of the constructions investigated to date does not compare with standard PU constructions and introduces excessive weight problems during full-scale board production. An example of this weight issue is seen in the studies by Michelena et al. and Correia et al. Both required large volumes of resin to wet-out the sustainable fabric used. This leads to complications during the hand layup and vacuum bagging process and dramatically increased the weight of their finished boards. Michelena et al. found their PU/flax/epoxy surfboard increased the weight by 25% compared to a standard PU/S-glass/polyester surfboard [23,24].

In the commercial sector, board manufacturers (shapers) have begun to produce surfboards with Expanded Polystyrene (EPS) foam cores as a result of issues with the supply of PU “blanks” (initial foam blocks specifically produced for the surfboard shaping process). These difficulties have arisen from closures to PU board blank manufacturing plants such as Clark Foam, which occurred due to increasing regulations on producing PU foam [22]. The EPS cores are now dividing the market due to their apparent performance benefits, such as greater buoyancy, improved strength, and reduced water permeation [26]. Despite the commercial popularity of EPS core surfboards, this material has not been fully investigated. All but one study includes EPS foam in their investigation. The only comparison study on a standard PU core board and an EPS core board is carried out by Oggiano and Panhuis [27]. This study outlines standard test methodologies to identify torsion stiffness and the damping response of manufactured boards. Their investigation presents the board comparison as a case study rather than determining benchmark material properties. A past study by the authors is the only experimental study to investigate these fundamental properties of EPS surfboard composites. This study reports the results of 4-point bend tests performed on a set of standard, high-performance, and sustainable surfboard construction utilising EPS foam cores [28].

The lack of EPS core surfboard literature is one of the several issues facing the understanding of the materials used in the sport. The lack of standardised testing methodologies further introduces variability between studies, with authors recognising this issue in the reliability of presented results. Furthermore, the use of hand layup techniques to manufacture test samples dilutes the repeatability of experimental data. Johnstone identified issues with this manufacturing technique, which are similarly seen in composite literature [29,30,31]. Other studies test the facing materials separately from the foam. While this investigates the strength of the facing materials, such testing completely ignores the bonding interaction between the facing and foam, which is critical to board performance, as poor bonding can lead to premature delamination. The agreement around the flexure properties being the most significant property to surfboard performance is not debated; however, there has been little investigation into other potential properties, which may impact the choice of board construction. Another missing component in the literature is an observational component highlighting the micro-scale material composition, which identifies the failure mechanisms of the materials.

This paper aims to provide a scientifically rigorous and standardised baseline for surf-centric sandwich composites. Through this investigation, the authors provide a range of mechanical properties for EPS/epoxy surfboard constructions. The mechanical tests include flexure, shear, fracture toughness, and tensile, with the subsequent mechanical properties presented and compared for various fabric reinforcements across two foam densities. The fabrics included in this study represent a cross-section of current surfboard compositions, with common base-level fabric layups (E-glass), high-performance fabrics (carbon fibre), and a sustainable alternative (recycled polyethylene terephthalate), which are all derived from choices made by current industry leaders. The production and test methodology described in this research introduces a robust and repeatable research process into surfing literature as an example for future researchers to follow and expand upon.

## 2. Materials and Methods

The EPS foam used to produce the samples was 5 mm thick, with (18.65 kg/m^3^) used as a medium (M) and (26.96 kg/m^3^) used for high (H)-density cores. The skin material and composite configurations used are shown in Table 1, with the Vee-Tek 2-part SURFSET flex epoxy resin matrix applied to all samples. 

Each composite panel had a foam surface area of 420 × 480 mm and was manufactured using a vacuum bagging process. Fabric preforms were cut 5 mm larger than the foam panels and each foam fabric layer was weighed separately. A flat aluminum plate long enough for three panels was used as a mold. The mold surface was cleaned with acetone, sprayed with release agent, and a tack tape was stuck around the edges of the plate. A release film was placed inside the taped perimeter, and three fabric preforms were laid on the film with a 20 mm gap between each (Figure 2a). The fabric weaves were then carefully wet-out by hand so as not to pull apart the fabric weave (a second layer was added for 2-ply composites). The foam was lightly wet with resin before being placed on the wet fabric for better bonding to the fabric layers. The top fabric weave was placed on the foam and carefully wet-out by hand (a second layer was added for 2-ply composites). After each laminate was uniformly wetted, a second release layer and bleed fabric were placed on each individual laminate. Finally, the vacuum hose was placed between two panels, and a plastic layer covering the entire build plate was stuck to the perimeter tact tape. The composite was then placed under atmospheric pressure and left to cure under pressure at room temperature for 48 h (Figure 2b). In each panel, test coupons were cut using an OMAX Waterjet 55100 machine (Figure 2c). Figure 2 illustrates the foam core sandwich panels at different stages of manufacturing, and a schematic of the vacuum bagging setup is shown in Figure 3.

The advantage of this process is an even wet-out of the composite at a low tooling cost, producing composites with consistent mechanical properties compared to hand layup [29]. The foam/fabric materials used in each composite were weighed before and post-lamination to determine the resin content of each composite and, accordingly, the fibre weight fraction (*W_f_*) from (1).
(1)$$W_{f}=\frac{panel\;weight-foam\;weight}{fabric\;weight}$$

### 2.1. Flexural Testing

The flexural properties of each composite construction were determined using 4-point bend testing following ASTM D7264. The test coupons were 12.8 mm wide and 120 mm long (Figure 4a). The span length of the coupons was 100 mm, with the load applied by a 50 kN Instron load frame at a rate of 5 mm/min at quarter span lengths (Figure 4b).

The max flexural stress (*σ_max_*) and modulus of elasticity (*E_f_*) were calculated using (2) and (3), respectively:(2)$$\sigma_{max}=\frac{3PL}{4bh^{2}}$$
(3)$$E_{f}=\frac{∆\sigma }{∆\varepsilon }$$
where *P* is the maximum load (N), *L* is support span (mm), *b* is coupon width (mm), *h* is coupon thickness (mm), Δ*σ* is the change in stress between two points (MPa), and Δ*ε* is the change in strain across the linear region of the engineering stress–strain curve (mm/mm). The engineering stress–strain curve is calculated from the built-in Instron encoder, as the low-strength material is unlikely to cause significant deflection in the testing system. The yield strength and strain are found from the engineering stress–strain curve using the 0.2% offset method. Material toughness (*U_T_*) at the point of ultimate stress is determined by the integration of the stress–strain curve. Six coupons were tested, with a minimum of five results contributing to the mean and standard deviation of the sample’s material properties. 

### 2.2. Fracture Toughness Testing

Fracture toughness experiments were performed using a 4-point bend fixture following ASTM D5045. A 50 kN Instron load frame was used to apply the load to the sample. The test coupons were 12 mm wide and 54 mm long, notched in the centre of one side (Figure 5a). The span length of the coupon was 50 mm, with the load being applied at a rate of 10 mm/min at ±20 mm from the centre (Figure 5b). 

Per ATSM D5045, *P_Q_* is found from the intersection of the force displacement (P-d) graph and a line drawn at an angle θ′, which is +5% of the θ of the linear region, as shown in Figure 6. If the failure point, *P_max_*, occurs after *P_Q_*, the fracture toughness (*K_IC_*) can be determined by the integration of stress–strain curves to the point of max stress. Eight coupons were tested, with a minimum of six results contributing to the mean and standard deviation of the sample’s material properties.

### 2.3. Iosipescu (V-Notch Shear) Testing

The shear strength of the samples was determined using Wyoming 2 fixture testing following ASTM D5379. A 10 kN Instron load frame was used to apply the load to the sample. The test coupons are 19.4 mm wide and 80 mm long, with notch dimensions as seen in Figure 7. The load was applied at a rate of 2 mm/min. 

The max shear stress (*τ_max_*) and shear modulus (*G_S_*) were calculated using (4) and (5), respectively:(4)$$\tau_{max}=\frac{P}{A}$$
(5)$$G_{S}=\frac{∆\tau }{∆\gamma }$$
where *P* is the maximum load (N), *A* is the sample cross-section area at the notch (mm^2^), Δ*τ* is the change in shear stress between two points (MPa), and Δ*γ* is the change in strain across the linear region of the engineering stress–strain curve (mm/mm). The engineering stress–strain curve is calculated from the built-in Instron encoder, as the low-strength material is unlikely to cause significant deflection in the testing system. The yield strength and strain are found from the engineering stress–strain curve using the 0.2% offset method. The shear toughness (*S_T_*) of the composite was determined by the integration of stress–strain curves. Six coupons were tested, with a minimum of five results contributing to the mean and standard deviation of the sample’s material properties.

### 2.4. Tensile Testing

The tensile strength of the samples was determined using screw side-action tensile grips following ASTM D3039. A 50 kN Instron mechanical tester was used to apply the load to the sample. The dogbone test coupon dimensions are given in Figure 8. The load was applied at a rate of 2 mm/min.

The ultimate tensile strength (*F_tu_*) and modulus (*E_T_*) were calculated using (6) and (7), respectively:(6)$$F^{tu}=\frac{P}{A}$$
(7)$$E_{T}=\frac{∆\sigma }{∆\varepsilon }$$
where *P* is the maximum load (N), *A* is the sample cross-section area at the middle of the sample (mm^2^), Δ*σ* is the change in stress between two points (MPa), and Δ*ε* is the change in strain across the linear region of the engineering stress–strain curve (mm/mm). The engineering stress–strain curve is calculated from the built-in Instron encoder, as the low-strength material is unlikely to cause significant deflection in the testing system. The yield strength and strain are found from the engineering stress–strain curve using the 0.2% offset method. Six coupons were tested, with a minimum of five results contributing to the mean and standard deviation of the sample’s material properties. 

## 3. Results and Discussion

### 3.1. Flexure Results

During flexure testing, all samples failed due to core wrinkling at the central rollers. This failure mode occurred underneath the top skin on the compression side of the coupon, indicating that the point of weakness in the sandwich structure is the compressive strength of the foam. This failure mode is common for foam core sandwich panels subjected to bending [19,32]. This failure is representative of critical damage, which occurs in surfboards and is colloquially known as “creasing”. The creasing of a surfboard can occur in many situations while surfing. Some of the expected conditions can include the waves breaking on the board, the surfer falling on the board after a wipeout, or when the surfer is attempting to perform expressive manoeuvres, such as aerials. Improving the flexural strength and material toughness of a surfboard is, therefore, a critical parameter of the surfboard composite, as breakages are an equipment longevity and sustainability concern, as well as a surfer safety consideration. In terms of a surfboard’s on-wave performance, flexural modulus is scrutinised more heavily. More flexible boards have been shown to have improved damping properties, which improve shock absorption in rough conditions and when completing expressive manoeuvres, such as aerials [27]. However, higher stiffness boards are considered to be more manoeuverable and sensitive to manoeuvres, owing to a faster feedback response to the surfer [26]. The mean results from the 4-point bend test indicate both the core and facing material have significant effects on the mechanical properties of the surfboard composite. Figure 9 compares the flexural properties of all composites, with (b–c) filtered by core density, where H-density and M-density are high- and medium-density, respectively. Full mechanical property mean values and standard deviations are available in the Appendix A.

All mechanical properties measured increased when the high-density foam was used for the core, apart from the strain and toughness for the single-ply E-glass composites, which saw a negligible change. This is attributed to the overall low strength of this material composition. The 1-EG facing produced the lowest overall strength and toughness, with a medium-density core and the second lowest with a high-density core. This supports qualitative investigations which determined surfers experience a short functional life for boards with current materials, as they are extremely susceptible to critical failure under common loading conditions [33]. As all other materials tested perform above this 1-EG threshold for strength and toughness, they are all viable materials for surfboard construction and are expected to extend the life of the surfboard under normal loading conditions.

The H-90CF laminate is both the stiffest and highest strength construction tested. The flexural modulus for this construction (2.28 GPa) was 58% higher than the next highest (H-2EG at 1.44 Gpa), and the flexural strength (4.92 Mpa) was 17% higher than the next strongest (H-2EG at 4.07 Mpa). The H-90CF composite also recorded high material toughness (0.0147 J/m^3^); however, the H-PET laminate was superior at 0.0153 J/m^3^. The difference between these two values was not of statistical significance. The relatively low deviation of the material toughness between fabrics indicates that this property is influenced more by core material rather than skin material. The PET fabric provides high strength but is the most flexible, with a modulus of 0.45 and 0.53 (M and H density, respectively). This is expected to be a result of the resin content of the PET composite samples, which recorded a lower *W_f_* than other samples owing to the fabric porosity and high fabric wet-out (Table 1). This supports the finding of previous studies, which indicated that sustainable fabrics were not direct replacements for standard material and incur resin volume issues.

Interestingly, when comparing the mid-range constructions, different fabric/foam combinations result in a similar flex. The H-1EG sample has only a 7% difference in modulus to the M-45CF construction. However, the M-45CF has a 16% increase in flexural strength, indicating the M-45CF construction has a greater resistance to creasing than H-1EG with similar flex behaviour. Likewise, the H-45CF laminate has only a 9% lower modulus than M-90CF but a 27% increase in strength. Similarly, a single layering of 90CF provides a 2% difference in flexural strength, the same as the 2EG samples when using a medium-density core. Such a reduction in fabric reinforcement, and thereby resin content needed for no significant reduction in strength, represents a clear advantage in delivering a lighter-weight board construction without sacrificing strength.

The results in Figure 9 indicate that combining changes to surfboard core material and reinforcing fabric can provide surfers with options for tailored stiffness. The higher-density core offers a broad range of stiffness, while medium density restricts this range. This quantitative comparison becomes useful to surfers and board designers assessing identical geometry boards. Higher-skilled athletes experimenting with custom boards can utilise both changes in fabrics and core material for precise calibration of a board’s flex pattern to the desired type of surfing, whilst lower-skilled surfers comparing boards only need to focus on boards with higher strength and toughness to prolong the life of the surfboard. 

### 3.2. Shear and Fracture Experiments

The shear and fracture experiments aid in the understanding of the crack development within the sandwich composites when damage is introduced to the board. The understanding of the resistance to crack propagation through the skin material is vital for surfboard manufacturers, as it is extremely common for surfers to continue to use their equipment with both minor and severe damage to the skin material. The fracture experiment results shown in Figure 10 are of interest as the intralaminar crack propagation will result in an irreparable board should the composite fail rapidly. Of the fabrics, the 45CF samples have the highest fracture toughness (0.0512 J/m^3^ for H, 0.0297 J/m^3^ for M) across the range of facing materials. This is a result of the fibre orientation in the double-bias weave. All other samples have plain weaves, which have half the fibres parallel and half perpendicular to the force application. Composites are known to be susceptible to crack propagation parallel to the fibre orientation, as was observed in the mechanical results. Interestingly, the foam density also had a significant impact on the fracture toughness of the materials. The high-density foam samples had an average difference of 51% compared to their medium-density counterparts. Hence, for purposes of fracture resistance, foam density in a surfboard is of careful consideration. Full mechanical property mean values and standard deviations are available in the Appendix A. The failure mode of the samples provides further insight into this relation.

The 1EG composites observed significant “brittle” failure through the entirety of at least one of the fibre facings as well as the foam, beginning from the propagation notch (Figure 11a–d). The 2EG samples saw similar brittle failure from the propagation notch; however, many of the samples remained intact, unlike the 1EG samples. The 2EG samples saw splits through the foam but only partially through the facing (Figure 11e,f). Observation of the fracture notch reveals that the fracture in the medium-density E-glass samples was affected by the air gaps between the foam bubbles. The fracture occurred between the outer surface of the foam bubbles for the M-density foam, as seen in Figure 11b; however, the H-density samples showed less evidence of bubble surfaces at the fracture point (d). 

For the 90CF and 45CF samples, the failure did not initiate from the notch but rather from the top side of the samples that were under compression. This indicates (similarly to the flexural experiments) that it is the compressive strength of the carbon that is the critical property for this facing material. The 90CF samples observed fibre breakages on the compression side; the fractures that occurred adjacent to the notch transferred to the notch through horizontal fibres once the notch depth was reached (Figure 12a,b). When comparing the two foams, M-90CF saw more evident creasing on this compression side. The H-density foam resists this creasing until the fabric itself cracks. Interestingly, the fracture toughness for M-90CF is lower than the M-2EG composite; however, H-90CF has higher fracture toughness than H-2EG, indicating that the higher foam density is more effective in utilizing the strength of the facing material. The 45CF composites also see failure occur due to compression forces creasing the foam and facing; however, for both foam densities, the facing itself remains intact (Figure 12c,d). This failure mode indicates the orientation of the fibres resists crack propagation at the notch and, due to the creasing, results in the highest toughness material. 

Finally, the PET samples were unique as they observed progressive “plastic” failure from the propagation notch, resulting in a higher toughness than the 1EG samples despite their low strength (Figure 13a,c). The PET sample also saw significant creasing on the compression side, which caused the facing to delaminate from the foam (Figure 13b,d); however, this creasing occurred without any fibres breaking, unlike the carbon facing.

For the Iosipescu experiments, there are typical fracture patterns that give acceptable results, as seen in Figure 14. 

The 1EG and 2EG samples failed compared to pattern B (Figure 15a,b), which is a clear indication of the low shear strength of the fibres, as it is most common for 90° uniaxial composites. The PET and 90CF samples failed more closely with pattern D (Figure 15c,d), which is a mixture of patterns commonly generated by the complex stress distribution seen in strong biaxial laminates. All four variations of these samples both failed with the foam and had initial cracking in the notched points. However, the 0° fibres underwent plastic deformation without breaking. Both patterns B and D produced acceptable results; however, the 45CF samples failed, akin to pattern F (Figure 15e). This failure was due to the foam crushing under the compressive stress from the fixture, rather than the facings under the shear loading. This was the only construction to fail in this pattern as, similar to the fracture experiments, the tow of this weave directly opposes the uniform shear stress contour created at the centre of the specimen. As a result, it can be assumed this material has the highest shear strength of the samples tested; however, the true magnitude has not been found. This is supported by past studies that established that axial composites are superior to orthogonal composites in shear loading [34].

**Figure 14 polymers-15-02703-f014:**
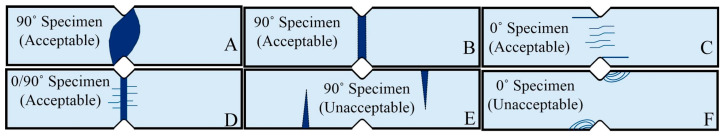
Typical acceptable and unacceptable fracture patterns of laminate and tow specimens in Iosipescu. Note: 0° is horizontal and 90° is vertical [35].

Regarding the acceptable samples, the EG composites report an 11% and 10% (1EG and 2EG, respectively) difference in shear yield strength and a 3% and 7% (1EG and 2EG, respectively) difference in the modulus compared to the M and H density foam. However, in the PET and 90CF samples, the shear strength and modulus both drop significantly when the density is lowered. For PET, the modulus decreased by 34%, and the yield strength decreased by 23% when switching to M-density foam. For 90CF, the modulus decreased by 24%, and the yield strength decreased by 29% when switching to M-density foam. This suggests each 0° toe of E-glass fibre has a shear strength comparable to the overall composite strength, whilst the 0° PET and 90CF fibre toes have a higher shear strength and remain intact during the loads where the foam and epoxy fail. Despite the low strength of the single-ply fibres, both densities of the 2EG samples have the highest shear strength of the acceptable results (3.11 MPa for H and 2.91 MPa for M), indicating that multiple fabric layers are important considerations in abating crack propagation through the surfboard’s skin materials. 

### 3.3. Tensile Experiments

Due to the compression face failing during flexural testing, it is important to understand the characteristics of each composite on the tension face. Whilst it is uncommon for a surfboard to be loaded in true tension while in use, the underside of the board planning on the water surface will be loaded in tension, which is an important consideration to overall material performance. The result of the tensile experiments is presented in Figure 16. Like the flexural results, the 90CF composites had the highest strength and modulus. The M-density results (which were the highest overall) recorded 142% higher strength and 127% higher modulus when compared to the baseline 1-EG samples. However, unlike the other experiments, it is clear the longitudinal tensile loading is less dependent on the density of the foam core as a result of the much higher strength and stiffness of the FRP facings compared to the EPS foam. This suggests variations between densities, as seen in the 2EG samples, are likely a result of fabric wet-out and interlayer adhesion rather than foam adhesion.

Of interest in these results is the 45CF-facing performance. The failure occurred via shearing between layers of the fabric weave (Figure 17), as the double-bias weave layers are stitched rather than interwoven. As a result, this construction recorded the lowest strength under this loading condition (3.36 MPa) despite its strength in the other conditions; however, the recorded stiffness was higher than the standard 1EG composites (as seen in other loading conditions). The tow orientation for the 45CF weave also diminishes the strength of this material in this loading condition. Contrary to the shear and fracture condition, the direction of the fabric tow does not align with the force direction in this condition. Therefore, the fibres do not directly oppose the force. This indicates that the 45CF construction’s maximum strength is mainly attributed to the resin matrix used and the adhesion quality during production.

## 4. Conclusions

The results reported in this paper investigate the mechanical behaviour of EPS core surfboards, which are gaining popularity in the surfing industry. The flexure, fracture, in-plane shear, and longitudinal tensile properties were investigated for a range of composites typically seen in modern surfboards. An optical assessment of the various failure modes is also presented. The effects of foam core density and fabric reinforcement on mechanical properties have been investigated. The following key conclusions are drawn from the research results:Ten different EPS core sandwich composites were manufactured via a vacuum bagging process. This process produced panels with highly consistent resin content for identical fabrics, for a low tooling cost.EPS foam density had the most significant influence on the flexural and fracture properties, where higher-density foam increased all but one metric across identical fabrics. It was also observed that there is less influence from the core density in the shear and tensile properties.The standard surfboard composite of a single layer of plain weave E-glass (1EG) was consistently one of the lowest composites for all the mechanical performance indicators. On the contrary, the mechanical properties of the plain weave carbon fibre (90CF) composite were frequently the highest of the fabric reinforcements measured. This construction is expected to be more suited for responsive boards for powerful surfing or surfing in dangerous conditions, where board durability is key.Microscopic analysis identified sudden brittle failure across the E-glass and carbon facings. However, PET facing observed severe plastic deformation and delamination, owing to its high resin content. The double-bias carbon facing (45CF) shear failure patterns were deemed unacceptable as there was evidence of core creasing due to Iosipescu fixture.The current standard material construction (H-1EG) was seen to have a stiffness of 0.84 ± 0.06 MPa. The results indicated surfboard manufacturers can utilise the M-45CF composite to obtain a similar board flex profile of 0.90 ± 0.04 MPa while improving the durability of the equipment, as the M-45CF composite increases the critical damage properties of flexural strength (+17%), material toughness (+107%), and fracture toughness (+156%), with only tensile strength decreasing (−57%). Shear toughness is not assessable; however, due to observed failure patterns, it is likely to be improved by the carbon composite.

The result trends heavily align with past composite literature, proving that the manufacturing methods are applicable for standardised rapid testing of advanced and sustainable surfboard constructions. The testing conducted indicates that standard flexural, fracture, and tensile testing are applicable for surfboard sandwich composites; however, careful consideration must be used for Iosipescu testing for sandwich composites with low-strength core materials. Further mechanical properties are recommended to be explored such as damping, impact resistance, and torsional stiffness.

The material properties presented in this study provide quantitative comparisons to reinforce the understanding of how surfboard materials perform for different loading conditions. The methods used in this study can be expanded to assess a wider range of surfboard composites for a lower production cost compared to producing a full surfboard for assessment, including a variety of material types, skin layer configurations, fabric weights, weave patterns, and core material options.

## Figures and Tables

**Figure 1 polymers-15-02703-f001:**
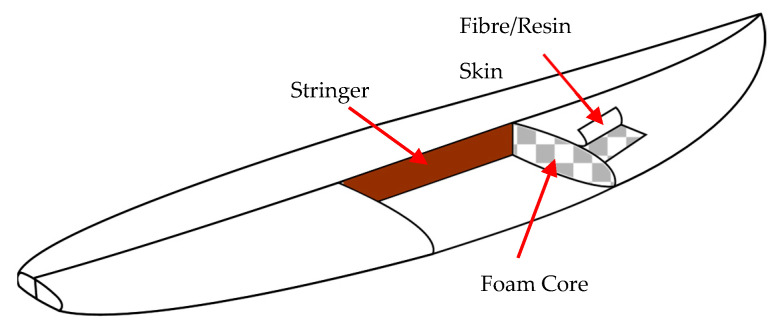
Typical surfboard sandwich composite.

**Figure 2 polymers-15-02703-f002:**
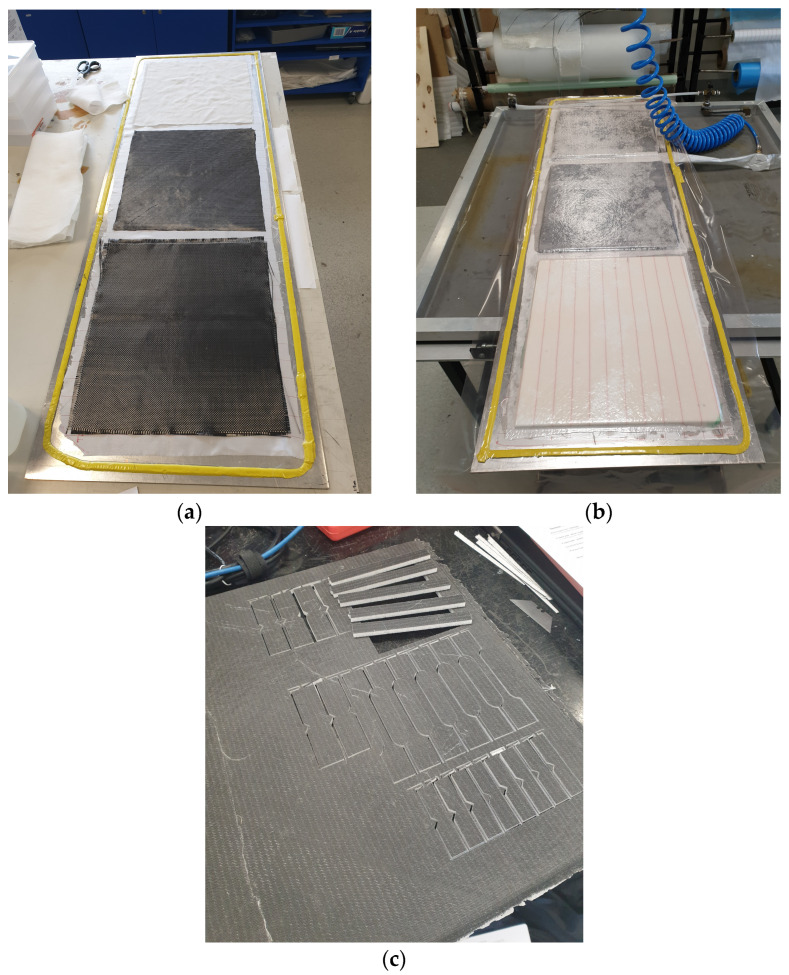
Stages of foam core sandwich panel fabrication: (**a**) initial prepped mold plate with dry fibres, (**b**) composite laminates at cure stage under vacuum, (**c**) finished composite laminate with samples cut out.

**Figure 3 polymers-15-02703-f003:**
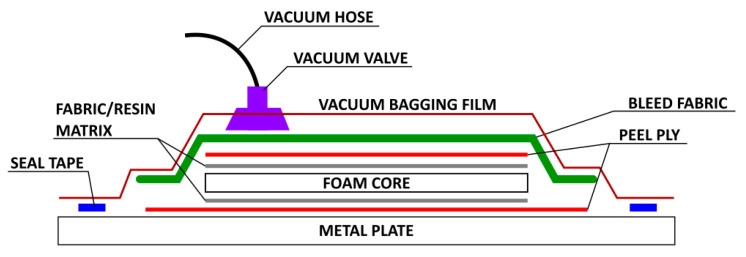
Vacuum bagging schematic for sandwich panel production.

**Figure 4 polymers-15-02703-f004:**
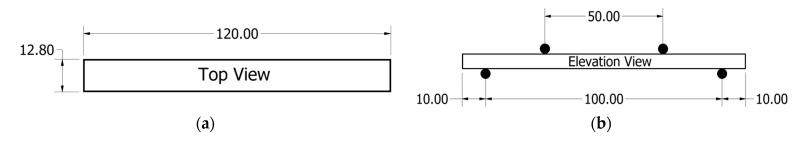
(**a**) Flexural sample cutting dimensions, (**b**) loaded schematic of flex sample. All dimensions are in mm.

**Figure 5 polymers-15-02703-f005:**
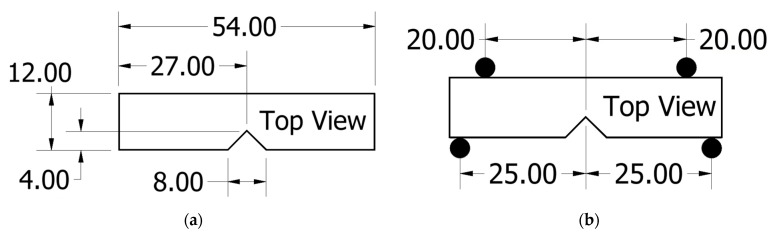
(**a**) Fracture toughness sample cutting dimensions, (**b**) loaded schematic of fracture sample. All dimensions are in mm.

**Figure 6 polymers-15-02703-f006:**
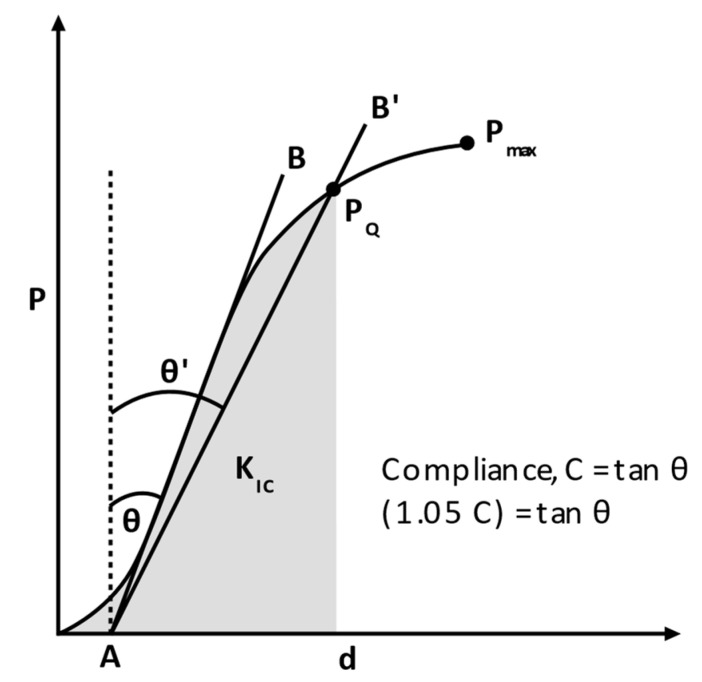
Determination of *P_Q_* from the force displacement graph.

**Figure 7 polymers-15-02703-f007:**
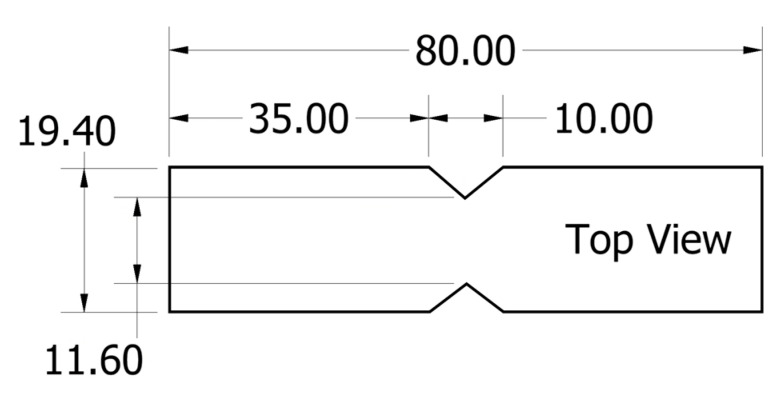
Iosipescu sample cutting dimensions. All dimensions are in mm.

**Figure 8 polymers-15-02703-f008:**
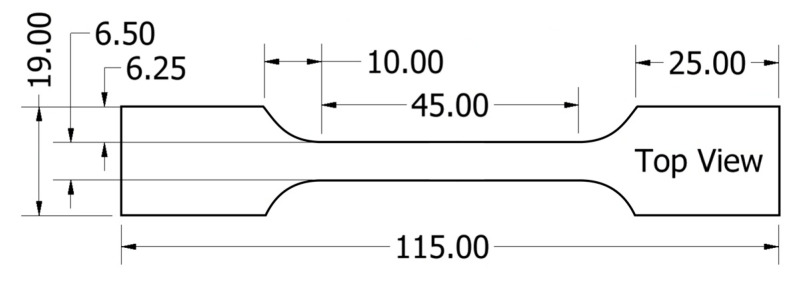
Tensile sample cutting dimensions. All dimensions are in mm.

**Figure 9 polymers-15-02703-f009:**
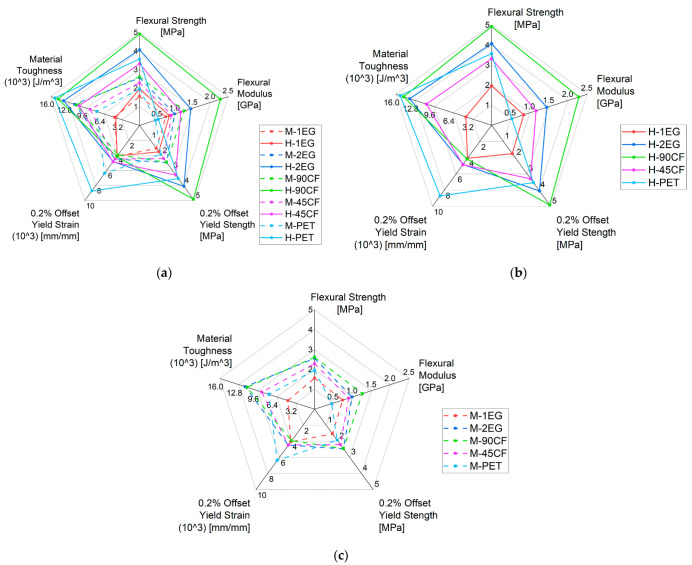
(**a**) Flexure experiment results, (**b**) results filtered by H-density, (**c**) results filtered by M-density [28]. See the Appendix A for mean values and standard deviations (Appendix A).

**Figure 10 polymers-15-02703-f010:**
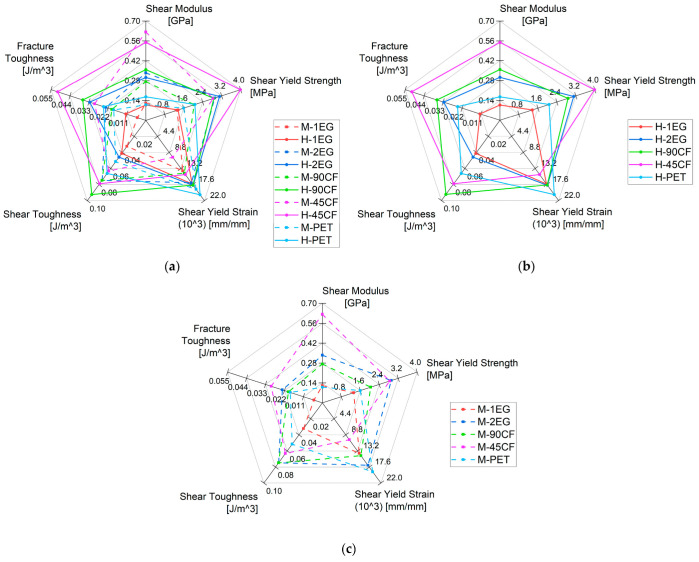
(**a**) Shear and fracture experiment results, (**b**) results filtered by H-density, (**c**) results filtered by M-density. See the Appendix A for mean values and standard deviations (Appendix A).

**Figure 11 polymers-15-02703-f011:**
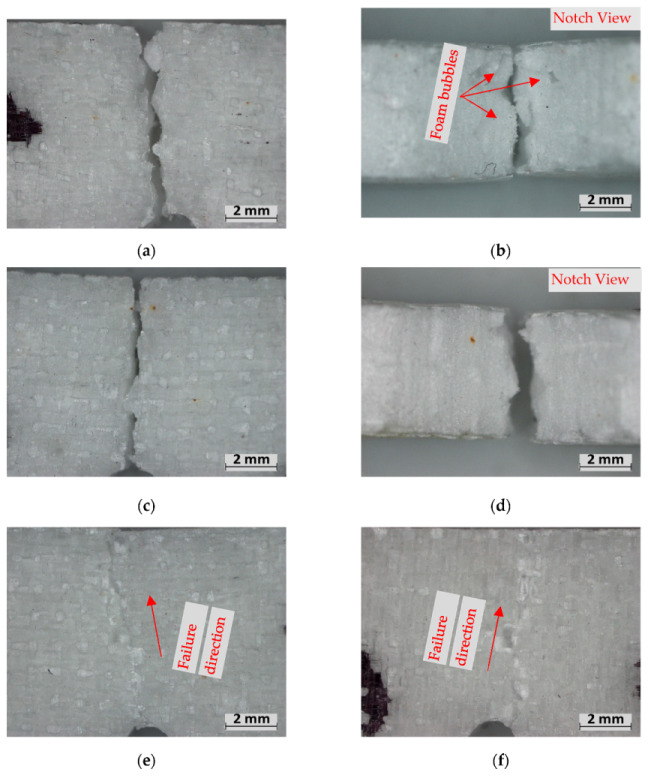
Fracture experiment failure: (**a**,**b**) M-1EG, (**c**,**d**) H-1EG, (**e**) M-2EG, (**f**) H-2EG.

**Figure 12 polymers-15-02703-f012:**
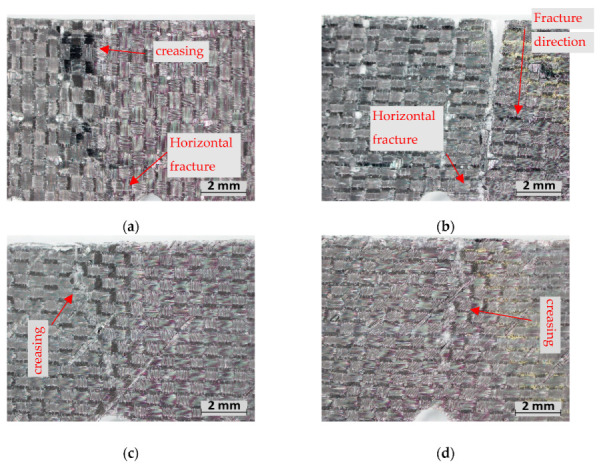
Fracture experiment failure: (**a**) M-90CF, (**b**) H-90CF, (**c**) M-45CF, (**d**) H-45CF.

**Figure 13 polymers-15-02703-f013:**
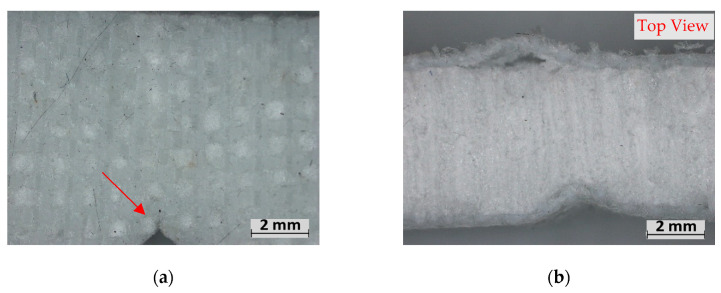
Fracture experiment failure: (**a**,**b**) M-PET, (**c**,**d**) H-PET.

**Figure 15 polymers-15-02703-f015:**
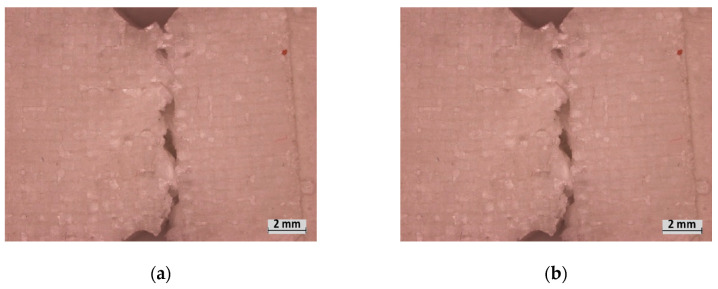
Iosipescu experiment failure: (**a**) M-1EG, (**b**) M-2EG, (**c**) M-PET, (**d**) M-90CF, (**e**) M-45CF.

**Figure 16 polymers-15-02703-f016:**
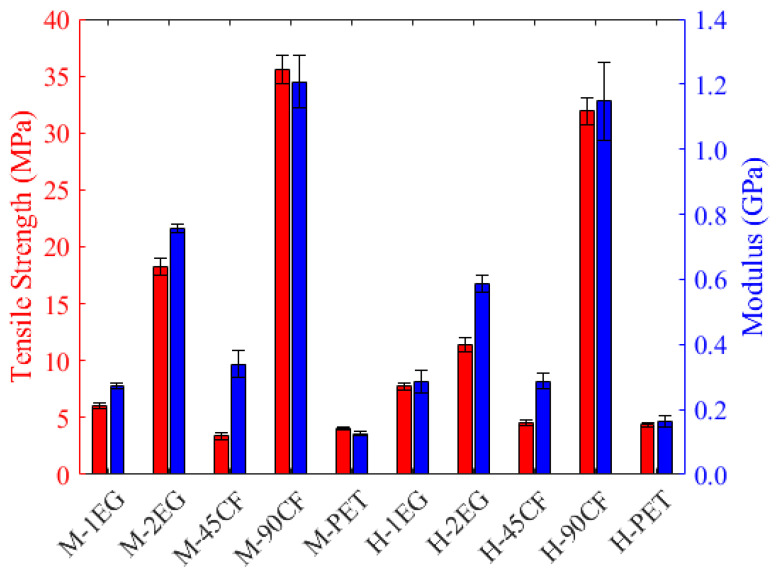
Tensile experiment results. See the Appendix A for mean values and standard deviations (Appendix A).

**Figure 17 polymers-15-02703-f017:**
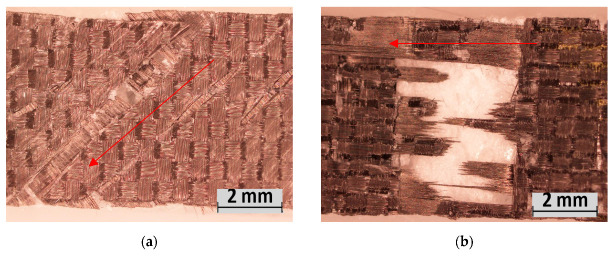
Tensile experiment failure: (**a**) M-45CF, (**b**) M-90CF.

**Table 1 polymers-15-02703-t001:** Experimental composite classification.

Foam	Fabric	Fabric Area Weight	Fabric Weave	No. Skin Layer	Code	Fabric Weight Fraction (W_f_)
High	E-Glass	135.62 gsm (4oz)	Plain 0/90	1-ply	H-1EG	0.47
Medium	E-Glass	135.62 gsm (4oz)	Plain 0/90	1-ply	M-1EG	0.46
High	E-Glass	135.62 gsm (4oz)	Plain 0/90	2-ply	H-2EG	0.54
Medium	E-Glass	135.62 gsm (4oz)	Plain 0/90	2-ply	M-2EG	0.51
High	Carbon Fibre	200 gsm	Plain 0/90	1-ply	H-90CF	0.50
Medium	Carbon Fibre	200 gsm	Plain 0/90	1-ply	M-90CF	0.52
High	Carbon Fibre	200 gsm	Double-Bias +45/−45	1-ply	H-45CF	0.49
Medium	Carbon Fibre	200 gsm	Double-Bias +45/−45	1-ply	M-45CF	0.50
High	Recycled Polyethylene Terephthalate (PET)	101.72 gsm (3oz)	Plain 0/90	1-ply	H-PET	0.26
Medium	Recycled Polyethylene Terephthalate (PET)	101.72 gsm (3oz)	Plain 0/90	1-ply	M-PET	0.27

## Data Availability

The data presented in this study is available in the Appendix A for this article.

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
