# Peer review of "An Experimental Investigation of the Mechanical Performance of EPS Foam Core Sandwich Composites Used in Surfboard Design"

_polymers, 2023, doi:10.3390/polym15122703_

Round 1

Reviewer 1 Report

Overall, the article is well written. The paper appears to be original/novel with potential to have good scientific impact in the sports/surfing community. However, I have several comments/suggestions that need to be addressed prior to recommending for publication. Please see the attached word document for comments.

Author Response

The authors would like to thank all reviewers for their time to read and assess the paper. The suggestions provided have improved the quality of the paper and we believe the changes to the manuscript as well as the justification of these changes have addressed the pitfalls and questions raised by the reviewers. Please see below the response to the comments.

The authors would like to note alongside the suggested revisions, a figure has been removed due to inability to obtain appropriate copyright permissions (originally Figure 2). Also, during editing the manuscript it was noticed that a paragraph discussing the Tensile results wasn’t included in the initial version sent and has been included in the revision (Page 17, Line 440-453). The paper has also been reviewed in full by English speakers as well as processed through a grammar and spelling checking program in an attempt to rectify any possible English errors. Should there be any more issues please point out the line they are located on and these will also be rectified immediately.

Thank you for your comments,

Sam Crameri

Reviewer 2 Report

The current paper adopts experimental methods to study the mechanical properties of sandwich composites. The flexural, shear and tensile properties are tested to evaluate the mechanical properties of composites. Some results may be important for promoting the application of sandwich composites. To further improve the quality of the paper, the following comments should be replied. 

1. It is recommended to provide some quantitative results or conclusions in the abstract section, especially in terms of mechanical performances and mechanism.

2. Fiber reinforced polymer composites have been widely used in many fields, such as aerospace, construction, automotive, and sports, due to their excellent properties and advantages. For example, CFRP has excellent mechanical properties, fatigue resistance, and corrosion resistance, but carbon fiber is relatively expensive. GFRP has good mechanical and corrosion resistance, and glass fiber is relatively cheap. The basic information and application backgrounds of FRPs should be added into the introduction section. Please review some latest relevant research below, such as Mechanics of Advanced Materials and Structures, 2023, 30(4):814-834. Composite Structures, 2021, 266:113864. Engineering Structures, 2023, 274: 115176.

3. Please provide some pictures of the sandwich composite prepared by the vacuum bagging method. In addition, please provide a further detailed description of the sample preparation and molding process.

4. The title spelling error in the flexural test of section 2.1 is suggested to be checked and corrected.

5. The clarity of Figure 5 is poor with smaller numbers, it is recommended to adjust it. In addition, please replace other unclear figures in the text.

6. Besides the basic mechanical performance testing, why not conduct some microscopic performance analysis? This is crucial for revealing the failure mechanism of the material

7. Please provide the relevant standard deviation for strength data in the figures.

8. From several mechanical performance indicators in figure 9, how to comprehensively evaluate the which material has the best flexural performance? Similar situations also apply to toughness, shear performance and tensile performance.

9. The conclusion should be further improved, including 3-4 key points. Some important preparation methods, performance variations and mechanism analysis are more desirable.

Need the minor change.

Author Response

(The authors gave the same response as above.)

Round 2

Reviewer 1 Report

The authors have addressed my comments. Paper is acceptable for publication. 

Reviewer 2 Report

It can be accepted.